# Attenuated viral strains of priority pathogens for potential use in controlled human infection model studies: A scoping review

David Oliver Hamilton[1]*, Victoria Simpson[1], Tilly Fox[1], Vittoria Lutje[1], Alain Kohl[2], Daniela M. Ferreira[1,3], Ben Morton[1]

**1** Department of Clinical Sciences, Liverpool School of Tropical Medicine, Liverpool, United Kingdom, **2** Centre for Neglected Tropical Diseases, Departments of Tropical Disease Biology and Vector Biology, Liverpool School of Tropical Medicine, Liverpool, United Kingdom, **3** Department of Paediatrics, Oxford Vaccine Group, University of Oxford, Oxford, United Kingdom

* oliver.hamilton@lstmed.ac.uk

## Abstract

### Background

There are several known pathogens and families identified as high risk for pandemic potential. It is essential to study these pathogens and develop medical countermeasures to mitigate disease prior to potential pandemics. Controlled human infection models (CHIMs) using attenuated viral strains may offer an efficient and safe way to do this.

### Objective

Our aim was to systematically examine the literature for attenuated, but replication competent, strains of Coalition for Epidemic Preparedness Innovations (CEPI) identified priority pathogens (Ebola, Lassa virus, Nipah virus, Rift Valley fever virus, chikungunya virus and Middle East respiratory syndrome-related coronavirus) that have been administered to humans.

### Design

A comprehensive literature search of multiple databases was performed by an information specialist. All search results were screened by two authors against inclusion/exclusion criteria from a pre-specified protocol. The primary outcome was confirmation that the administered viral strain could subsequently be recovered from participants. The secondary outcome was attenuated virus safety.

### Results

Our searches yielded 13078 results and 5998 articles remained for screening after removing duplicates and animal studies. Subsequently, 351 articles were selected for

**Data availability statement:** All relevant data are within the manuscript and Supporting information file.

**Funding:** The author(s) received no specific funding for this work.

**Competing interests:** The authors have declared that no competing interests exist.

full text review and nine were included for data extraction. Four distinct attenuated strains were identified across two priority pathogens – TSI-GSD-218 and VLA1553 for chikungunya virus and MP-12 and hRVFV-4s for Rift Valley Fever virus. Attenuated virus was recovered for each strain except hRVFV-4s. There were no major safety concerns for these identified strains in Phase 1–3 studies.

## Conclusions

We have identified three attenuated viral strains that may be amenable to development into novel CHIMs for two priority pathogens. Of these, VLA1553 for chikungunya is a licenced and commercially available vaccine product suitable for use in CHIM. There is a research gap for the creation of new attenuated mutants that could be utilised in CHIM for other priority pathogens.

## Author summary

There are several families of viruses that scientists predict are most likely to cause a future pandemic, such as coronaviruses or Ebola. Studying these viruses ahead of time might mean we have vaccines or drugs already available before an outbreak occurs.

Human challenge studies involve exposing healthy volunteers to germs that might make them sick and can be an efficient way of testing vaccines or treatments. It wouldn't be safe to do that for many illnesses that could cause a pandemic. However, it may be possible to use weakened versions of these viruses instead. We have performed a thorough search of scientific papers to look for candidates of weakened viruses to see if we could use them like this. The most promising is a version of the chikungunya virus that is currently used as a vaccine.

## 1. Introduction

Viruses with epidemic and pandemic potential risk destabilising international economies and social order; and could cause mass illness and deaths [1]. This threat is increased by anthropogenic climate change; land-use ecosystem changes, increased human and livestock populations; and potentially due to bioterrorism [2]. The Coalition for Epidemic Preparedness Innovations (CEPI) have targeted the manufacture of safe vaccines, therapeutics, and diagnostics within 100 days of identification of an emerging pandemic [3]. Modelling data suggests that over 8 million deaths and $14 trillion could have been saved if this 100-day target had been met during the COVID-19 pandemic [4].

Whilst the causative organism of the next viral pandemic may be entirely novel (so called "Disease X" [2]), there are several known pathogens or families identified as

high risk for pandemic potential, particularly *coronaviridae, orthomyxoviridae and filoviridae* [1,5–7]. CEPI have defined nine diseases prioritised for development of medical countermeasures (MCMs) prior to a potential pandemic: Ebola virus disease (EVD); Lassa; mpox; Nipah; Rift Valley fever (RVF); chikungunya (CHIK); COVID-19; Middle East respiratory syndrome (MERS); and "Disease X" [5,8]. Development of MCMs against these viruses is essential to improve pandemic preparedness [1,5]. As occurred in the COVID-19 pandemic, knowledge gained from studying one of these viruses may also expedite MCM development if a novel pandemic virus were to emerge from a related family or genus [9,10].

Standard Phase 1–3 efficacy studies to develop experimental MCMs may be impossible due to limited or absent participants to enrol into trials prior to disease outbreaks. Controlled human infection models (CHIMs) may offer an efficient solution to this critical dilemma [11]. Traditionally, CHIMs involve deliberate exposure of an infectious dose of a pathogen to carefully selected volunteers [12]. CHIMs may significantly accelerate the development of MCMs in a safe, efficient and cost-effective way [13–15], for example by up- or down-selecting potential agents to ensure that only the most promising candidates are progressed to pivotal efficacy trials [14,16]. CHIM derived data has recently been used for regulatory approvals for VaxChora for travellers [17] and the World Health Organization (WHO) pre-qualification of the use of a typhoid conjugate vaccine (TypbarTCV) in endemic regions [18,19]. It is recognised that CHIMs may support the emergency use of an investigational vaccine in a pandemic scenario [20] and therefore could be a crucial tool in pandemic preparedness [4,16].

Given the inherent risks from some of these priority pathogens, CHIMs that incorporate wild type viral infection would be unethical [21–23] and therefore novel approaches to study design are required. Previous studies have used attenuated versions of a disease-causing pathogen to mitigate these risks [14,21,24], usually repurposing an attenuated strain designed as a live-attenuated vaccine. This is sometimes termed 'pseudochallenge' [14]. The approach has been successful in CHIM with several other diseases, most notably dengue virus [25,26], as well as tuberculosis [27], influenza [28], rotavirus, and poliovirus [14]. Given the lack of precedent with CHIMs in these diseases and the presumed heterogeneity of studies, we have conducted a scoping review to systematically examine the literature for attenuated strains of CEPI priority pathogens that have already been administered to humans. This will identify candidates that may be developed into novel CHIMs to facilitate trials for MCMs.

## 2. Methods

The objectives, eligibility criteria and methods for this scoping review were specified in advance and published in a prospectively registered protocol on the Open Science Framework (https://osf.io/nu3bf/). The scoping review was conducted according to methodology from the JBI Manual for Evidence Synthesis [29] and incorporates the PRISMA Extension for Scoping Reviews (PRISMA-ScR) checklist [30] (see S1 File).

### 2.1. Search strategy and eligibility criteria

A comprehensive literature search was last performed on 24th February 2025 in the Cochrane Central Register of Controlled Trials (CENTRAL, published in the Cochrane Library), MEDLINE (via OVID), Embase (via OVID), Science Citation Index (Web of Science), CAB Abstracts & Global Health (Web of Science) databases. We also searched the WHO International Clinical Trials Registry Platform (ICTRP; apps.who.int/trialsearch/) and ClinicalTrials.gov (https:// clinicaltrials. gov/ct2/home) for trials in progress. The full search terms used are included in the S2 File. There were no restrictions on language, region, date, participant demographics, or publication status. Additionally, references of all identified reviews were also hand-searched to identify potential additional eligible studies.

### 2.2. Study selection

Inclusion criteria were determined *a priori*: adult humans (≥ 18 years old); deliberately exposed to a near-whole-genome, attenuated version of any of the following priority viruses: Ebolavirus; Lassa mammarenavirus; Nipah virus (NiV);

Rift Valley fever phlebovirus; chikungunya virus (CHIKV); or Middle East respiratory syndrome–related coronavirus (MERS-CoV).

For a successful CHIM, it is essential to confirm infection and clearance with microbiological or virological techniques [27,31]. Therefore, we excluded studies of mutations that preclude representative viral replication: chemical inactivation; irradiation; sub-unit vaccines; virus-like particles; recombinant viruses with genetic material from another virus such as vesicular stomatitis virus or chimpanzee adenovirus; and mRNA or DNA vaccines. Studies without primary data, such as editorials or systematic reviews were also excluded. COVID-19 has already been developed into a well-established CHIM [32,33] and was therefore not included in this scoping review. CEPI lists a hypothetical, unknown "Disease X" as a priority pathogen for vaccine development. An unknown disease is not amenable to human challenge so this priority pathogen group was excluded. Some strategies for the rapid development of a CHIM for a new pathogen in a pandemic scenario are discussed elsewhere and are outside the scope of this review [16,34–36]. Mpox was designated a CEPI Priority Pathogen online after completion of the scoping review protocol and was not included [8].

Two investigators (DOH and VS) independently screened titles and abstracts using Rayyan (https://www.rayyan.ai/) [37]. The first 25 title and abstracts were screened together as a pilot to ensure consistency. No automated tools were used. All abstracts deemed potentially eligible by either author proceeded for full text review by both authors, recorded using Microsoft Excel (Microsoft, WA, US). Discrepancies of full text studies were resolved by discussion or by a third investigator (BM).

## 2.3. Data extraction

The primary outcome was confirmation the administered viral strain could subsequently be recovered from participants. The secondary outcome was safety of the mutant viruses. Other outcomes were narratively summarised where reported, namely: author; year; institution; mutation from wild-type; study phase; dosage; sample size; comparator; adverse events (AE)/serious adverse events (SAE); follow-up length; and availability and regulatory requirements.

The full framework for data extraction is presented in S3 File. This was developed iteratively with input from authors expert in CHIM development as the search developed. Data was extracted by a single-author (DOH), recorded using Microsoft Excel, and checked by a second (VS). A risk of bias assessment was conducted by a single author (DOH). We used the original Cochrane Collaboration Tool [38] for randomised studies. This tool was the most appropriate because the outcome of interest in our review (viraemia) was not the primary outcome of the studies evaluated and this tool provides a general risk of bias assessment rather than against a particular outcome. Non-randomised studies were assessed using the ROBINS-E tool [39].

## 2.4. Data synthesis

Some attenuated strains were investigated in more than one study. In those cases, the methodology and results of those studies are presented together (Table 1 and S3 File). We have provided a descriptive and quantitative (where appropriate) summary for each identified attenuated strain. A background for each pathogen is also presented prior to the description of any identified attenuated strains.

## 3. Results

The literature search resulted in 13,078 studies (n = 6242 for EVD, n = 2517 for MERS-CoV, n = 1591 for CHIKV, n = 1197 for Rift Valley fever virus (RVFV), n = 911 for Lassa virus (LV) and n = 620 for NiV). We first removed 3653 duplicates and then, as per our protocol, removed 3427 articles found via search-terms that referenced non-human primates, with the option they could be included later if very limited human data was found (this step was not subsequently required). Thus, 5998 studies remained for title and abstract screening. Of these, 351 manuscripts were selected for full text review and nine were included for data extraction (see Fig 1). Table 1 describes the characteristics of included studies and each

**Table 1. Results and characteristics of studies included after full-text review, with further detail in S3 File.**

| Author & Year | Candidate name & Pathogen | Study Phase | Number exposed | Rate of recovery of attenuated virus | Method for detection of attenuated virus | Adverse event incidence | Serious adverse events | Suitable for CHIM |
|---|---|---|---|---|---|---|---|---|
| McClain 1998 [40] | TSI-GSD-218 CHIKV | Phase 1 | 55 | 36.8%* | Amplification in cell culture | Overall AE rate not reported | Nil | Yes |
| Edelman 2000 [41] | TSI-GSD-218 CHIKV | Phase 2 | 59 | Not reported | N/A | Overall AE rates not reported | Nil | Yes |
| Hoke 2012 [48] | TSI-GSD-218 CHIKV | Phase 1 | 51 | Not reported | N/A | Overall AE rates not reported | Nil | Yes |
| Wressnigg 2020 [46] | VLA1553 CHIKV | Phase 1 | 120 | Not reported** | RT-qPCR | 73% | 0.8% (1/120) | Yes |
| Schneider 2023 [42] | VLA1553 CHIKV | Phase 3 | 3082 | Not reported | N/A | 62.5% | 1.5% (46/3082) | Yes |
| McMahon 2024 [43] | VLA1553 CHIKV | Phase 3 | 408 | Not reported | N/A | 72.5% | 1.2% (5/408) | Yes |
| Pittman 2016a [45] | MP-12 RVFV | Phase 1 | 69 | 16.3% (7/43) | Direct plaque assay and nucleic acid amplification | Overall AE rates not reported | Nil reported | Yes |
| Pittman 2016b [47] | MP-12 RVFV | Phase 2 | 19 | 26.3% (5/19) | Blind passage of plasma on Vero cells | 89.5% | Nil | Yes |
| Leroux-Roels 2024 [44] | hRVFV-4s RVFV | Phase 1 | 60 | 0% (0/60) | RT-qPCR | Overall AE rate not reported | Nil | No- given no recoverable virus |

AE = adverse event, CHIKV = chikungunya virus, CHIM = controlled human infection model, N/A = not applicable, RT-qPCR = quantitative reverse transcription polymerase chain reaction, RVFV = Rift Valley fever virus. *- numerator/denominator not presented (presumed 7/19) ** - Wressnigg 2020 presents mean cohort genome copy equivalents only.

pathogen is described separately, with further detail in S3 File. Table 2 presents the pipeline of attenuated viruses across the pathogens. All attenuated viruses identified in this scoping review had been developed as part of a search for an effective live-attenuated vaccine.

Of the nine included studies, five were randomised controlled trials [40–44], two were randomised controlled Phase 1 trials with a non-randomised safety or confirmatory cohort [45,46] and two were non-randomised interventional studies [47,48]. Five of the nine studies investigated for recovery of attenuated virus [40,44–47]. S4 File presents risk of bias assessments for the included studies. Only one study was found to be at high risk of bias [45].

### 3.1. CHIKV

This alphavirus is transmitted by *Aedes* mosquitoes and can cause explosive epidemics, particularly in urban areas [23,60,61]. At the time of writing, there is an active epidemic in La Réunion with >47,000 cases reported since August 2024 [62]. CHIKV is endemic to several continents including Africa, Asia, the Americas and, more recently, southern Europe [63,64]. There is international concern that climate change will increase the spread of CHIKV by expanding the habitat for its vector into previously infection-naïve populations [63,64]. Chikungunya disease is characterised by fever,

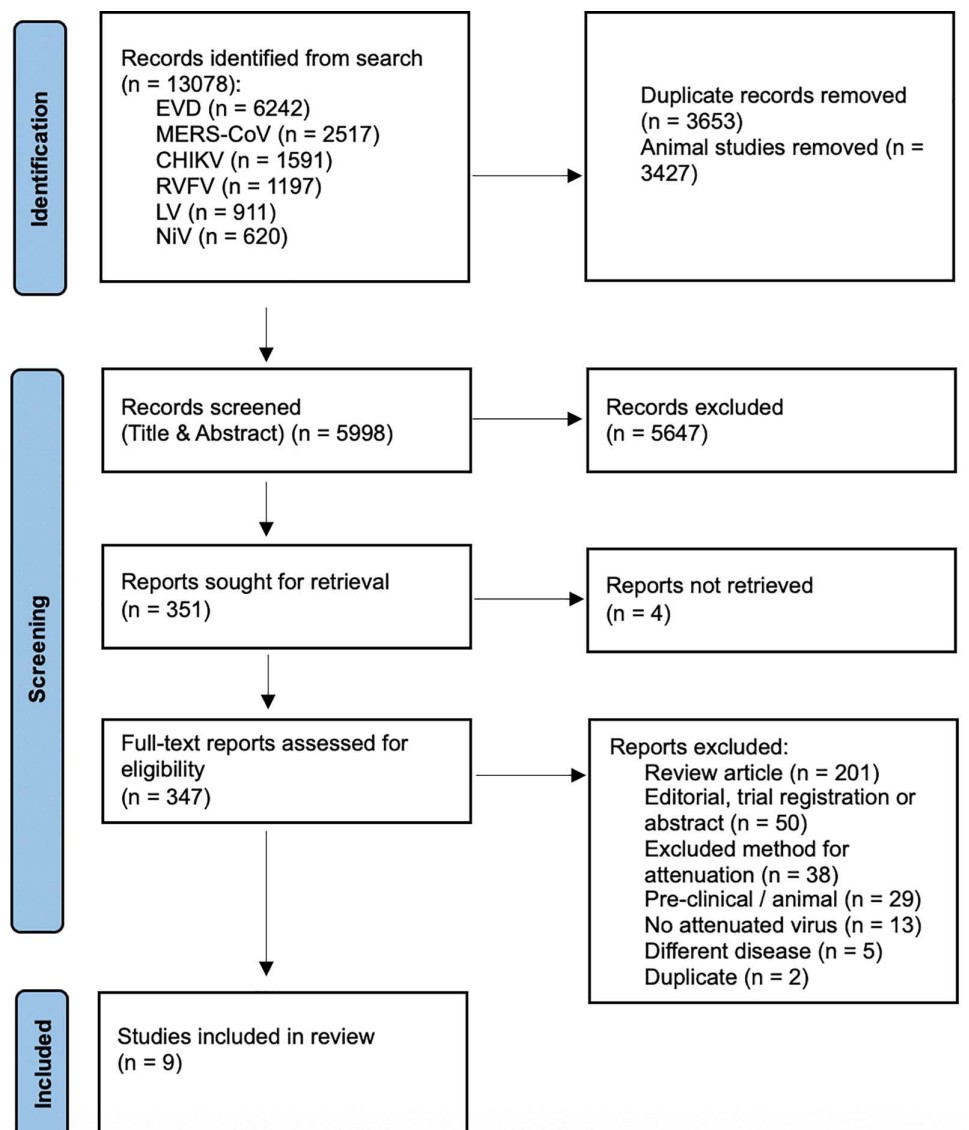

**Fig 1. PRISMA Flow Diagram for the Scoping Review process.** CHIKV = chikungunya virus, EVD = Ebola virus disease, LV = Lassa virus, MERS-CoV = Middle East respiratory syndrome-related coronavirus, NiV = Nipah virus, RVFV = Rift Valley fever virus.

malaise and arthralgia with a case-fatality rate around 0.1%, although this can be higher in older or co-morbid adults [23]. The disease is associated with a high level of long-term morbidity as it can lead to a chronic, debilitating arthritis, which accounts for a substantial global socioeconomic burden [65]. There are no licenced antiviral treatments against this infection [66]. There are two recently licensed vaccines against CHIKV: Ixchiq, a live-attenuated vaccine using strain VLA1553 that is discussed in detail below [67]; and Vimkunya, a virus-like particle (VLP) vaccine [68,69]. There are multiple attenuated CHIKV strains that have been used in pre-clinical or animal models [70–79], although only two have proceeded to use in humans, VLA1553 and TSI-GSD-218.

**3.1.1. VLA1553.** VLA1553 (initially termed Δ5nsP3), was developed as a single-shot live-attenuated vaccine. First described by Hallengärd et al [80], attenuation was achieved by deleting a large part of the gene *nsP3* encoding the

**Table 2. Pipeline of attenuated virus controlled human infection model candidates for Coalition for Epidemic Preparedness Innovations (CEPI) identified priority pathogens (excluding COVID-19, mpox and "disease X") across the translational development pipeline, organised by strains in most advanced stage of study.**

| Pre-clinical | Phase 1 | Phase 2 | Phase 3/ Post-marketing |
|---|---|---|---|
| LV [49,50] | EVD [51] | RVFV [47] | CHIKV [42,43] |
| MERS-CoV [52–57] | | | |
| NiV [58,59] | | | |

CHIKV = chikungunya virus, EVD = Ebola virus disease, LV = Lassa virus, MERS-CoV = Middle East respiratory syndrome-related coronavirus, NiV = Nipah virus, RVFV = Rift Valley fever virus.

non-structural replicase complex protein nsP to minimise the risk of reversion. The candidate is based on the 2004–2005 epidemic La Réunion strain, produced in Vero cells and purified by centrifugation, ultrafiltration, batch-chromatography, and sucrose gradient centrifugation. The mutated virus was shown to be genetically stable, safe and protective in a mouse model [80], and later in a non-human primate model [81].

Wressnigg et al. conducted a Phase 1 study in 120 healthy adults using a single-shot of three escalating doses of VLA1553 [46] (S3 File). Systemic solicited AEs were experienced by 52.5% (63/120) of participants and 10.8% (13/120) of participants experienced severe related adverse events. There were no adverse events of special interests (AESIs) and there was one unrelated SAE (S3 File). Following the first dose, a transient increase in viral RNA was detected in all cohorts by quantitative reverse transcription polymerase chain reaction (RT-qPCR) using a hydrolysis probe and primers specific to the CHIKV gene *nsP1*, which peaked at Day 3 and resolved by Day 14. Urinary shedding of attenuated virus was only detected at one time point in one participant. Of note, in 94 participants administered repeat immunisation 6–12 months later, attenuated virus was not detectable by RT-qPCR and only 3.2% (3/94) of participants experienced targeted systemic symptoms solicited by investigators.

The safety of the candidate was demonstrated in a large double-blind, multicentre, placebo-controlled, randomised Phase 3 trial when given to 3082 participants (of 3093 randomised) [42]. AEs were experienced by 62.5% (1926/3082) of participants who received VLA1553, with the majority headache, fatigue and myalgia, although 18.0% (554/3082) experienced arthralgia. SAEs were reported in 1.5% (46/3082) of participants exposed to VLA1553 compared to 0·8% (8/1033) of participants in the placebo arm, although only two of the SAEs were deemed related to the vaccine (one mild myalgia in a 58-year-old patient with known fibromyalgia leading to a five-day hospitalisation for investigation and one admission for presumed syndrome of inappropriate antidiuretic hormone secretion and atrial fibrillation in a 66-year-old patient following a fever on day 11). Recovery of attenuated virus from the blood of participants was not attempted in this study. A second Phase 3 study of 408 participants examining three lots of VLA1553 further confirmed the safety of this attenuated virus, although this study did also not attempt to recover attenuated virus [43]. The use of VLA1553 as a live-attenuated vaccine (licenced as Ixchiq) has been approved in Canada, the European Union and the UK and tens of thousands of doses have been administered globally [67,82–84]. However, use has recently been paused in adults aged over 65 in Canada and the UK due to concern regarding post-marketing safety reports, including three deaths, in this subpopulation [84–86]. Having initially licenced Ixchiq for use in the USA, the Food and Drug Administration has since suspended this licence for all adults due to these safety concerns [84].

**3.1.2. TSI-GSD-218.** An attenuated version of CHIKV, later termed TSI-GSD-218 and also known as CHIK 181/clone 25, was created by Levitt et al. in 1985 from a Thai strain serially passaged in primary green monkey kidney cells and later in Medical Research Council (MRC)-5 cells in an attempt to develop a live-attenuated vaccine [87]. The attenuation is only mediated by two point mutations in the E2 glycoprotein [88].

Phase 1 studies were conducted by the US Military but have been only partially published [40,48]. In the cohort of alphavirus-naïve participants a Phase 1 study [40], 36.8% (presumed 7/19) had detectable viraemia upon amplification in cell culture for 1–2 days from Day 4–8, although none could be directly plaqued from serum. The overall number of AEs is not reported although there was a low number of typical solicited AEs and the authors state that these were not significantly different from the placebo cohort. Summary data from previously unpublished Phase 1 studies are presented by Hoke et al. with no significant safety concerns reported [48].

In a Phase 2 study, 73 healthy volunteers were recruited to a randomised, placebo-controlled trial of vaccination with TSI-GSD-218 (4:1 vaccine to placebo) [41]. Related AEs were similar across both cohorts (32% [19/59] vs 29% [4/14]) with two severe related AEs and 8% (5/59) experiencing temporary arthralgia compared to 0% in the placebo arm. The levels of viraemia are not reported in this study. Our search did not identify any studies involving TSI-GSD-218 that were actively recruiting since the year 2000, reportedly due to "changes in assessment of threats to military operations" alongside "anticipated difficulties" in demonstrating efficacy [48].

### 3.2. RVFV

This bunyavirus is transmitted by multiple mosquito species between humans and ruminants [89]. The resultant RVF disease is a major One Health and economic threat as it can cause epidemics of fatal disease in both humans and livestock [90]. RVFV is endemic to sub-Saharan Africa and the Arabian Peninsula. There is concern it could spread further due to spill-over events from imported infected herds or via the increasing global reach of its vectors [91]. RVFV causes a wide-spectrum of human disease including encephalitis, hepatitis, retinitis and viral haemorrhagic fever, fatal in around 20% of hospitalised patients [90,92,93]. There are currently no licenced vaccines or antivirals against RVFV for use in humans. There are multiple attenuated versions of RVFV in the scientific literature, often developed as a live-attenuated vaccine for animals [94–106], and based on our search, we identified two that have progressed to use in humans [44,45,47], strains MP-12 and hRVFV-4s.

**3.2.1. MP-12.** First developed in the 1980s by the US military, MP-12 is an attenuated strain of RVFV created for both human and veterinary use by performing 12 serial plaque passages of the Egyptian strain ZH548 through MRC-5 cells in the presence of 5-fluorouracil [107]. The attenuation is based on mutations in all of the S-, M- and L-segments of the virus [89], providing some protection against reversion to wild-type [102]. MP-12 has been shown to be generally safe and immunogenic in ruminants [108–111] and non-human primates [112,113], whilst also causing a low level viraemia. However, it was shown to be potentially teratogenic in early pregnancy and may cause a hepatitis in young animals [92,101].

Two Phase 1 studies of MP-12 were previously unpublished but later summarised by Pittman et al. in their Phase 1 dose escalation and route comparison study as part of the assessment of MP-12 as a live-attenuated vaccine [45]. Firstly, four participants received undiluted MP-12 ($10^{4.4}$ plaque forming units [PFU]) as a subcutaneous (SC) injection. All four participants developed a mild-moderate transaminitis which resolved without sequelae. Attenuated virus was recovered from one participant using nucleic acid amplification. A further 22 participants were randomised to placebo or four different dilutions of MP-12 SC (S3 File). Transaminitis, raised lactate dehydrogenase and creatinine kinase (CK) are reported but the authors summarise the vaccine as "remarkably safe" [45]. In the published data of the Phase 1 study, performed in 1996 and published in 2016, 56 healthy volunteers were randomised to various doses of MP-12 either SC or intramuscular (IM) (S3 File) [45]. No SAEs and no significant solicited symptoms are reported, however there was one self-resolving Grade 4 transaminitis. Three Grade 4 rises in CK are also reported, although the authors comment that these were likely related to military exercises performed by participants. Virus could be recovered by direct plating of serum in one participant and in a further six by tissue culture amplification and *in situ* detection via Enzyme-Linked Immunosorbent Assay (ELISA) (7/43 [16.3%] exposed participants).

A Phase 2 study by the same group administered $10^5$ PFU IM to 19 healthy volunteers [47]. Solicited AEs, including headache, fever and injection site pain, were frequent but well tolerated (S3 File). No significant related biochemical

abnormalities and no SAEs are reported. Assessment of viraemia was performed on plasma and buffy coat specimens collected daily for 14 days post exposure using both plaque assay and blind, double passage on Vero cells. No viraemia was detected by direct plaque assay in any participant. Viraemia detection using blind, double passage on Vero cells was detectable in 5/19 (26.3%) of participants (1–4 isolates per subject, between Day 4 and Day 9). There was no reversion to wild-type in recovered virus between participants. We identified no studies that administered MP-12 to humans after 2008. There are reports that MP-12 vaccine candidate was paused due to cold-chain and BioSafety Level-3 requirements; and liver toxicity concerns [114].

**3.2.2. hRVFV-4s.** hRVFV-4s is a mutant RVF virus created by splitting the glycoprotein precursor gene to produce a four-segment virus [115]. It has been demonstrated to be safe in mice [115], ruminants [116] and non-human primates [117], whilst causing no detectable viraemia. A Phase 1 study of 75 participants in a placebo-controlled (3:1), dose-escalation study of hRVFV-4s demonstrated that the attenuated virus was well tolerated with only mild-moderate solicited symptoms and no related Grade 3–4 symptoms or SAEs [44]. Importantly, no vaccine viral RNA was detected via RT-qPCR in any blood, urine, saliva or semen samples from participants at numerous timepoints (days 0, 1, 3, 7, 14, 28, and 180).

## 3.3. Ebolavirus species

Viruses within the *Filoviridae* are RNA viruses which cause viral haemorrhagic fever (VHF) with high morbidity and mortality and is transmitted person-to-person via direct contact with infected bodily fluids [118]. It has caused devastating epidemics across West and Central Africa, most notably the 2013–2016 West African epidemic which caused 11,325 deaths [119]. There are four species of the genus Ebolavirus that cause disease in humans: Sudan ebolavirus (SUDV), Bundibugyo ebolavirus, Taï Forest ebolavirus (TAFV), and Zaire ebolavirus (EBOV) [120]. The SUDV and EBOV species have historically caused most epidemics [119].

Two vaccines have been licenced in the USA or Europe: rVSVΔG-ZEBOV-GP (Ervebo) [121] and Ad26.ZEBOV+heterologous MVA-BN-Filo boost (Zabdeno/Mvabea) [120]. Only Mvabea may provide coverage against non-EBOV species as it expresses EBOV, SUDV & Marburg virus glycoproteins plus TAFV nucleoprotein [122]. There are two antiviral treatments licenced for EBOV: atoltivimab-maftivimab-odesivimab (Inmazeb) and ansuvimab (Ebanga) [123]. Attenuated whole-genome Ebola is uncommon in the pre-clinical literature [118,124–128], although one, EBOVΔVP30, has progressed to both non-human primate [129] and Phase 1 human study [51].

**3.3.1. EBOVΔVP30.** Halfmann et al. developed a mutant that lacks the viral protein (VP) 30 gene, known as EBOVΔVP30 [118]. VP30 is an essential transcription factor for EBOV and hence this virus is replication-deficient outside of Vero cells expressing VP30 *in trans*. It has been shown to be genetically stable, morphologically indistinct from wild-type and safe in a mouse model [126]. The mutant was further inactivated with hydrogen peroxide when transferred to a non-human primate model [129]. According to the trial registry, in 2019, a Japanese group enrolled 15–30 healthy human volunteers to a Phase 1 study using EBOVΔVP30 (named 'iEvac-Z') [51]. This study has not yet been published, although a conference abstract describes "a strong safety profile in humans" [130]. As there are no further published data to assess, the study did not meet our pre-specified inclusion criteria for this scoping review. There are press reports that state a second study was due to open in Sierra Leone in 2024 [131].

## 3.4. LV

LV is an arenavirus endemic to West Africa [132]. It causes Lassa fever, a VHF responsible for around 5000 deaths per year [133], for which there are no licenced vaccines or therapeutics. LV may be transmitted by rodents or person-to-person via infected bodily fluids [49]. We identified no example of whole-genome attenuated LV that has been administered to non-human primates or humans. There are however recent examples of live attenuated viruses created by reverse genetics and administered to guinea pigs in an attempt to create a novel live-attenuated vaccine [49,50]. The fact

that Argentine haemorrhagic fever virus, another arenavirus causing VHF, has a licenced live-attenuated vaccine in current use [134] gives further credence to the idea that a safe attenuated LV mutant may be feasible.

There is an example of an attenuated virus, ML29, rationally created by reassortment of LV and Mopeia virus (MOPV), an attenuated relative of LV, that has been administered to non-human primates [135,136]. This carries the replication machinery of MOPV and expresses major antigens of LV, however as this is genetically distinct from LV itself and only results in a low, transient viraemia, it did not meet pre-specified criteria for inclusion in our scoping review [136].

### 3.5. NiV

NiV is a henipavirus that has caused sporadic outbreaks throughout Asia. It is spread by bats, livestock or human-to-human transmission and therefore of high pandemic potential [137,138]. It can cause a lethal encephalitis with a high case-fatality rate and has no licenced therapy or vaccination [138,139]. Whilst there are attenuated whole-genome versions of NiV in a pre-clinical setting [58,59], we identified no studies that have progressed to human use.

### 3.6. MERS-CoV

MERS-CoV is a coronavirus similar to severe acute respiratory syndrome coronavirus (SARS-CoV) and COVID-19, which can cause a fatal pneumonia with a high case-fatality rate [140]. It is spread by dromedary camels or by person-to-person contact [141]. MERS-CoV is largely contained within in the Arabian Peninsula, although it has caused a large outbreak in South Korea when imported by a returning traveller [142]. There are no current licenced vaccines or antivirals against MERS-CoV [143]. Whilst there are multiple attenuated whole-genome versions of MERS-CoV in pre-clinical use [52–57], we identified no studies that have progressed to human use.

### 4. Discussion

We have conducted a rigorous scoping review to identify existing attenuated strains of six CEPI priority pathogens that have been used in humans that could potentially be progressed into novel human challenge models. We have identified four such strains across two priority pathogens, although only three strains (TSI-GSD-218 [40,41,48] and VLA1553 [42,43,46] of CHIKV and MP-12 [45,47] of RVFV that produce the required virological response necessary for a CHIM [27]. The final identified strain, hRVFV-4s of RVFV, does not cause viraemia in pre-clinical settings [117] nor was virus detected in humans despite robust RT-qPCR testing [44]. There was insufficient published data to assess the attenuated EBOV strain EBOVΔVP30, although based on the pre-clinical data [129], it is unlikely that it would cause detectable viraemia for use as a primary endpoint within a CHIM. Three of the CEPI priority pathogens, NiV, MERS-CoV and LV, have no existing attenuated strains administered to humans.

Of the three identified strains where attenuated virus may be recovered from the host, only VLA1553 has progressed to Phase 3 testing and licensure in the form of the live-attenuated vaccine Ixchiq [42,43,67,83]. VLA1553 has been demonstrated to be tolerable and safe and appears to produce consistent detectable viral RNA with an immunological response similar to natural infection [46,83,144–146]. The use of attenuated virus raises concerns about reversion to wild-type, however to date there has been no reports of this for VLA1553 and the deletion of a large part of the gene *nsP3* renders this unlikely [80]. The lack of detectable virus and markedly reduced solicited adverse events seen following re-exposure 6–12 months later can be interpreted as a proof-of-concept that VLA1553 could be useful as a challenge agent investigating other MCMs against CHIKV, with effects on attenuated viral RNA and symptoms as outcome measures. However, the safety concerns with VLA1553 that have been identified post-licensure in older or co-morbid patients demonstrate the importance of careful participant selection in a future hypothetical CHIM [84,85]. It is also unknown if VLA1553 would be detectable if administered SC or intra-dermally, which would imitate a mosquito bite more closely. Nevertheless, VLA1553 remains a more promising candidate for development into a CHIM for CHIKV than TSI-GSD-218 given the greater clinical experience and more stable mutation (S3 File).

Typically, live-attenuated viruses are developed as early vaccine candidates and have a long history of safe use, for example, in yellow fever, smallpox and polio [147]. However, such candidates may be discontinued as potential vaccines if they are unacceptably reactogenic, even if they are safe and immunogenic. Reactogenicity is of lower concern in a human challenge agent if meticulous informed consent is obtained; symptoms are mild/moderate and participants are closely monitored. There is precedent with dengue for progressing an abandoned live-attenuated vaccine candidate (rDEN2Δ30) into a successful attenuated CHIM [25,148]. The efficacy of the TV003 vaccine that was demonstrated in the CHIM by Kirkpatrick et al. [25] was later replicated in Phase 3 field trials [149]. We postulate that this model could be imitated with VLA1553 in CHIKV to test novel therapeutics or additional vaccine candidates that may be more suitable for pregnant, older or immunocompromised patients than VLA1553. A safe CHIM for CHIKV would be an important advance due to: the limitations of animal models [150]; the lack of a universally-accepted correlate of protection [23,151]; the difficulties in conducting field tests due to unpredictable and often short-lived outbreaks [23,66]; and the lack of licenced antiviral [23,66].

Similarly, it may be possible to develop MP-12 into a CHIM for RVFV. It has been shown to be well tolerated and safe in Phase 1 and 2 studies when administered both SC and IM to around 100 healthy volunteers [45,47] and is conditionally licenced in animals [114]. However, the recovery rate of MP-12 in participants is low at 16.3-26.3%, which would necessitate a large sample size in a hypothetical CHIM [45,47], although these rates are based on techniques of direct plaque assay, nucleic acid amplification or blind passage through Vero cells. More work would be required to determine the attack rate using contemporary diagnostic assay techniques. In order to progress MP-12 into a CHIM, a method of recovering the strain by RT-qPCR would need to be developed. Whilst we identified no human studies that were actively recruiting since 2008, there is suggestion in the literature that this was still being developed as a live-attenuated vaccine in 2020 by the Sabin Vaccine Institute [102].

There are a number of other attenuated RVFV strains used in veterinary practice but none appear suitable for development into CHIM, either due to safety concerns in the case of the Smithburn strain [89] or the lack of viraemia in the Clone-13 strain [102]. There are also several so-called "next-generation" MP-12 strains in pre-clinical development, such as r2segMP12, DDVax and RVax-1 [89,101]. These are strains that have been attenuated through reverse genetics of the NSs protein, a major virulence factor, however their safety in humans or their ability to cause a viraemia is currently unknown [89] and further study would be required before incorporation into a CHIM.

During the COVID-19 pandemic, SARS-CoV-2 CHIM studies were performed [32,33] after development of a robust ethical framework [16,152]. These studies represent the first time a CHIM had been used in an active pandemic [32]. COVID CHIM studies provided unique data on the underlying pathophysiology [33], transmission of the disease and the accuracy of lateral-flow testing [32]. However, recruitment to these studies still took over a year from the initial onset of the pandemic [31], by which time vaccines had been developed and licensed using traditional (accelerated) routes. It is recognised that creating an attenuated virus may take at least a year of study during a pandemic [36], and therefore we have searched for strains that could be used in advance of an epidemic or pandemic to model the efficacy of candidate MCMs in the pipeline [153].

Our scoping review has several strengths. To our knowledge, it is the first review to systematically investigate existing attenuated viral strains of priority pathogens that could be re-purposed into a CHIM. We involved an information specialist to develop our search strategy and adhered closely to both our pre-specified protocol and recognised methodology for a scoping review [29]. In common with all scoping reviews, our findings are dependent on the quality of the included studies leading to some limitations. Whilst all individual candidate strains had at least one study reporting detectable attenuated virus in blood, only four of the nine included studies reported these rates precisely [40,44,45,47]. There was also incomplete reporting of overall adverse event rates, with five studies not reporting these data precisely [40,41,44,45,48]. Furthermore, there were notable incomplete or delayed publications on early phase work with strains TSI-GSD-218 (CHIKV) and MP-12 (RVFV) by the US military, which undermines confidence in the safety of progressing those strains into CHIM [40,45,48]. We made a pragmatic decision to focus the scoping review on six viruses on the CEPI priority pathogen list [5],

rather than the 23 viruses on the WHO "list of emerging pathogens for a potential future pandemic" [1], due to logistical capacity. However, the same methodology and data collection framework could be applied to other viruses with pandemic potential in future work.

In conclusion, there are three attenuated viral strains of two CEPI priority pathogens, CHIKV and RVFV, that have been administered to humans that cause detectable attenuated viral RNA in blood and may therefore be amenable to development into a novel CHIM. Of these, VLA1553 for CHIKV is a licenced and commercially available vaccine product and therefore suitable for immediate use in CHIM. There is a research gap for the creation of new attenuated mutants that could be utilised in CHIM for other priority pathogens, but the availability of reverse genetics systems and sound knowledge of proteins such interferon antagonists, or codon deoptimization strategies could allow the future development of such attenuated viruses.

## Supporting information

**S1 File. PRISMA-Scr Checklist.** From: Tricco AC, Lillie E, Zarin W, O'Brien KK, Colquhoun H, Levac D, et al. PRISMA Extension for Scoping Reviews (PRISMAScR): Checklist and Explanation. Ann Intern Med. 2018;169:467–473. https://doi.org/10.7326/M18-0850.
(DOCX)

**S2 File. Search strategy.**
(DOCX)

**S3 File. Expanded Table of Data Extracted from all included studies.**
(DOCX)

**S4 File. Risk of Bias Assessment.**
(DOCX)

## Author contributions

**Conceptualization:** David Oliver Hamilton, Tilly Fox, Daniela M. Ferreira, Ben Morton.

**Data curation:** David Oliver Hamilton, Victoria Simpson, Vittoria Lutje.

**Formal analysis:** David Oliver Hamilton.

**Methodology:** David Oliver Hamilton, Tilly Fox, Ben Morton.

**Supervision:** Daniela M. Ferreira, Ben Morton.

**Writing – original draft:** David Oliver Hamilton, Ben Morton.

**Writing – review & editing:** Victoria Simpson, Tilly Fox, Vittoria Lutje, Alain Kohl, Daniela M. Ferreira.

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
