## [Decision Letter · Decision Letter 0]

29 Oct 2025

Response to Reviewers
Revised Manuscript with Track Changes
Manuscript

Shaden Kamhawi

co-Editor-in-Chief

Paul Brindley

co-Editor-in-Chief

**Journal Requirements:**

**Comments to the Authors:**

**Please note that one review is uploaded as an attachment.**

**Reviewers' comments:**

**Key Review Criteria Required for Acceptance?**

**Methods**

-Are the objectives of the study clearly articulated with a clear testable hypothesis stated?

-Is the study design appropriate to address the stated objectives?

-Is the population clearly described and appropriate for the hypothesis being tested?

-Is the sample size sufficient to ensure adequate power to address the hypothesis being tested?

-Were correct statistical analysis used to support conclusions?

-Are there concerns about ethical or regulatory requirements being met?

Reviewer #1: Standard structured literature review, hampered by only partial access to old publications where results were not appropriately published.

Reviewer #2: Objectives of the study are clearly articulated. As it was a literature review, there was no testable hypothesis and no study population. Sample size and statistical analyses were not applicable. There are no concerns about ethical or regulatory requirements. The study design is appropriate for the stated objective of systematically reviewing literature.

Reviewer #3: The objectives of the study care articulated with a clear testable hypothesis stated in this review paper.

The Mat and methods are clearly described and appropriate for the hypothesis being tested.

**Results**

-Does the analysis presented match the analysis plan?

-Are the results clearly and completely presented?

-Are the figures (Tables, Images) of sufficient quality for clarity?

Reviewer #1: As a literature review the manuscript is sufficient. A table for each of the pathogens studied with key study data would be useful, certainly more so than Figure 2 which could go to supplement. Some more space could go to the AE description

Reviewer #2: The analysis is aligned with the analysis plan and the results are clearly and completely presented. Figures are of sufficient quality for clarity.

Reviewer #3: the analysis presented matches the analysis plan. The results are clearly and completely presented. The legends of the figure have to be corrected

**Conclusions**

-Are the conclusions supported by the data presented?

-Are the limitations of analysis clearly described?

-Do the authors discuss how these data can be helpful to advance our understanding of the topic under study?

-Is public health relevance addressed?

Reviewer #1: I am left underwhelmed by the lack of thoughtful and more extensive consideration in the discussion of what uses should/could an attenuated virus be put to, how to engineer it, manufacture, test in a pre-clinical fashion. Now with sophisticated genetic tools and cell culture it would seem to be important to highlight the potential of modern approaches to this important issue. Deficiencies in the reporting of older studies by the US military should be pointed out.

Reviewer #2: The conclusions are supported by the data presented. There was no description of limitations of the analysis. The authors do discuss how these data could be helpful in the development of controlled human infection models that may accelerate future development of vaccines against CEPI Priority Pathogens, which would be highly relevant to public health.

Reviewer #3: Conclusions are supported by the data/figures/tables presented

**Editorial and Data Presentation Modifications?**

Reviewer #1: (No Response)

Reviewer #2: Please consider the following minor revisions:

Lines 49-50: "...VLA1553 for chikungunya is in the most advanced stage of development..." I consider this to be an understatement because VLA1553/Ixchiq is licensed in the European Union, therefore I would consider it's clinical development to be essentially complete, rather than at an "advanced stage". Consider instead, "...VLA1553 is a licensed and commercially available vaccine product suitable for use in a controlled human infection model.

Line 293: Consider clarifying here that Ixchiq is the commercial name for VLA1553. As currently written, it isn't completely clear these are the commercial and development code names for the same vaccine.

Lines 301-302: As was done later for some of the other attenuated viruses, consider adding a comment about the risk of reversion of VLA1553 to a more virulent strain based on the nsP3 deletion. It isn't obvious why this deletion would "eliminate" the risk of reversion.

Line 308: Please clarify if subjects in this clinical study received 3 different dose levels (e.g. low, medium, and high doses) or if each subject received 3 immunizations (and if so, at what interval?).

Line 312: I apologize for being somewhat pedantic, but viremia refers to detection of replication competent viruses in blood, for example by plaque assay. qRT-PCR only detects viral RNA, not the presence of replication-competent viruses. Consider rewording: "...a transient increase in viral RNA after the first dose was detected by qRT-PCR..."

Line 315: "Shedding of attenuated viruses was only detected at one time point..." Given that CHIKV is transmitted by an insect vector, it isn't clear what "shedding" refers to. Does this mean attenuated viruses were detected in serum? Please clarify.

Line 325: Consider adding details on the SAEs deemed related. Given that this virus is being proposed for use in a CHIM, safety is a paramount concern.

Line 326: "Viraemia was not reported in this study" Please clarify if this means the authors of the study did not report on this outcome, or if viremia was assayed for, but not observed.

Line 329: VLA1553, not "VLA1153".

Lines 329-330: the license of Ixchiq has been revoked by the US FDA, therefore this section needs to be updated to reflect the current state.

Lines 339-340: Consider adding a comment about risk of TSI-GSD-218 reverting to virulence, given that attenuation is mediated by only two point mutations. Given that this strain is being proposed for use in CHIMs, stability of the attenuation is a prime concern.

Line 350: "...randomised 4:1 to TSI-GSD-218..." I assume this means 4 vaccine to 1 placebo, but please clarify.

Lines 408 and 542: The reported attack rate of 26.3% for MP-12 is relatively low compared to strains used for other CHIMs and would have significant implications on the number of subjects needed to enroll in order to sufficiently power a CHIM study testing a vaccine candidate. Consider adding comment on this in the Discussion of the suitability of MP-12 for use in a CHIM.

Lines 439-440. The statement that there are no licensed antiviral medications for EBOV is incorrect. There are two monoclonal antibody therapies licensed by the US FDA for EBOV: Inmazeb and Ebanga. These should be mentioned here. The reference cited for this statement (#117 Jacob et al. Nat Rev Dis Primers) was published in February 2020, whereas the two mAb therapies were approved in October and December 2020, respectively. Consider citing a more recent reference that describes Inmazeb and Ebanga.

Reviewer #3: Specific comments

Figure 1 : Legends : CHIKV = chikungunya should be Chikungunya virus, EBOV, Ebola virus, LV, Lassa virus, MERS should be MERSV Middle East Respiratory Syndrome virus, RVFV = Rift Valley fever virus. MERSV has to be corrected in the figure

Table 2 Legend : same remarks as Figure 1 legends

**Summary and General Comments**

Reviewer #1: (No Response)

Reviewer #2: The authors have conducted a systematic literature review to identify potential attenuated virus strains of several CEPI Priority Pathogens that could be used in controlled human infection model (CHIM) studies. This comprehensive review does an excellent job of summarizing the various studies conducted with these virus strains. Hopefully this review will stimulate the development of these and other attenuated strains for use in future CHIM studies.

General comment: the description of the strains focused on their replication in clinical studies, which is appropriate as the primary consideration for a CHIM strain. However, please consider adding comments about the immunological responses elicited by these strains, if known, as this could be compared with the immunological responses to virulent wild-type strains and thus serve as an important indicator of how useful and valid the attenuated strains would be as challenge strains.

Reviewer #3: Although this systematic review focuses on attenuated viral vaccines, the discussion could have included a table listing other promising vaccines that have already been administered to humans for each of the priority pathogens, in order to provide an overview of vaccines in clinical trials in humans for these different viruses.

Specific comments

Figure 1 : Legends : CHIKV = chikungunya should be Chikungunya virus, EBOV, Ebola virus, LV, Lassa virus, MERS should be MERSV Middle East Respiratory Syndrome virus, RVFV = Rift Valley fever virus. MERSV has to be corrected in the figure

Table 2 Legend : same remarks as Figure 1 legends

PLOS authors have the option to publish the peer review history of their article (what does this mean? ). If published, this will include your full peer review and any attached files.

**Do you want your identity to be public for this peer review?** For information about this choice, including consent withdrawal, please see our Privacy Policy .

Reviewer #1: **Yes: ** James McCarthyh

Reviewer #2: No

Reviewer #3: No

**Figure resubmission:****Reproducibility:** To enhance the reproducibility of your results, we recommend that authors of applicable studies deposit laboratory protocols in protocols.io, where a protocol can be assigned its own identifier (DOI) such that it can be cited independently in the future. Additionally, PLOS ONE offers an option to publish peer-reviewed clinical study protocols. Read more information on sharing protocols at https://plos.org/protocols?utm_medium=editorial-email&utm_source=authorletters&utm_campaign=protocols

---

## [Editor Report · Decision Letter 1]

15 Dec 2025

Dear Dr Hamilton,

We are pleased to inform you that your manuscript 'Attenuated viral strains of priority pathogens for potential use in controlled human infection model studies: A scoping review' has been provisionally accepted for publication in PLOS Neglected Tropical Diseases.

Best regards,

Gregory Gromowski

Academic Editor

Michael Holbrook

Section Editor

Shaden Kamhawi

co-Editor-in-Chief

Paul Brindley

co-Editor-in-Chief

---

## [Editor Report · Acceptance letter]

Dear Dr Hamilton,

We are delighted to inform you that your manuscript, "Attenuated viral strains of priority pathogens for potential use in controlled human infection model studies: A scoping review," has been formally accepted for publication in PLOS Neglected Tropical Diseases.

Best regards,

Shaden Kamhawi

co-Editor-in-Chief

Paul Brindley

co-Editor-in-Chief
